# Intestinal Microbes and Hematological Malignancies

**DOI:** 10.3390/cancers15082284

**Published:** 2023-04-13

**Authors:** Yinghong Zhu, Qiaohui Yang, Qin Yang, Yanjuan He, Wen Zhou

**Affiliations:** 1Haihe Laboratory of Cell Ecosystem, State Key Laboratory of Experimental Hematology, National Clinical Research Center for Geriatric Disorders, Department of Hematology, Xiangya Hospital, Central South University, Changsha 410008, China; 2Key Laboratory for Carcinogenesis and Invasion, Chinese Ministry of Education, Key Laboratory of Carcinogenesis, Chinese Ministry of Health, Cancer Research Institute, School of Basic Medical Sciences, Central South University, Changsha 410008, China; 3NHC Key Laboratory of Human Stem and Reproductive Engineering, School of Basic Medical Science, Central South University, Changsha 410205, China; 4Department of Hematology, Third Xiangya Hospital, Central South University, Changsha 410013, China

**Keywords:** intestinal microbes, hematological malignancies, clinical features, initiation, progression

## Abstract

**Simple Summary:**

Gut microbes, which constitute the most complex and essential micro-ecosystem in the body, are known as the second genome of the human body. The relationships of micro-ecosystems have been increasingly studied over the last few years; it has been shown that human metabolism and immunity may be altered in the absence of microbial balance, which not only affects the initiation and progression of malignant malignancies, but affects tumor treatment response and tumor treatment-related complications as well.

**Abstract:**

Hematological malignancies are diverse, with high malignancy characteristics, poor prognoses, and high mortality rates. The development of hematological malignancies is driven by genetic factors, tumor microenvironment factors, or metabolic factors; however, even when considering all of these factors, one still cannot fully estimate the risk of hematological malignancies. Several recent studies have demonstrated an intimate connection between intestinal microbes and the progression of hematological malignancies, and gut microbes play a primary role in the initiation and progression of hematological tumors through direct and indirect mechanisms. Thus, we summarize the correlation between intestinal microbes and hematological malignancies’ onset, progression, and therapeutic effect in order to better understand how intestinal microbes affect their initiation and progression, especially in leukemia, lymphoma, and multiple myeloma, which may provide potential therapeutic targets for improving the survival of patients with hematological malignancies.

## 1. Introduction

In terms of the source of cells, namely, bone marrow, blood, and lymph nodes, hematological malignancies can be divided into four types: myeloid neoplasms, lymphoid neoplasms, histiocyte tumors, and mast cell tumors, all of which belong to the immune system [1,2,3]. The majority of hematological tumors are malignant, with a poor prognosis and high mortality rate. In the process of disease, hematological neoplasia patients are at risk of complications such as infection and sepsis, which lead to multiorgan dysfunction, increasing treatment difficulty, and can even pose a tremendous threat to patient survival.

Human gut microbes make up the most complex and essential micro-ecosystem in the body. On one hand, the number of microbial cells in the human intestinal tract accounts for 78% of the total quantity of microbial cells in the body, and the weight of human intestinal microbes is up to 1 kg; on the other hand, it is estimated that there are about 1000 species in human intestinal flora. Additionally, the number of human intestinal microbial genes is about 3.3 million, 150 times the number of human genes (20,000 to 25,000), known as ‘the second genome’ of human health [4]. As the essential micro-ecosystem in the body, gut microbes maintain a long-term mutualistic relationship with the host through fermenting dietary fiber, defending against pathogens, and synthesizing short-chain fatty acids (SCFAs) or beneficial metabolites. Moreover, the immune system’s maturation and host function are also susceptible to the homeostasis of gut microbes. The role SCFAs play in intestinal homeostasis has been confirmed, and it is further correlated with tissues and organs beyond the intestines through blood circulation [5]; SCFAs not only signal through binding to cognate G-protein-coupled receptors on endocrine and immune cells in the body, they also induce epigenetic changes in the genome through effects on the activity of histone acetylase and histone deacetylase enzymes [6]. The interaction between the mammalian intestine and its resident gut microbiota is delicate; the gut establishes tolerance with commensal microbes, maintains control over pathobionts, and protects the functions of intestinal epithelial barrier by preventing microbial overgrowth. In turn, microbiota in the gut modulate and regulate the immune system in order to promote tolerance within the host. Any disruption to this exquisite equilibrium might cause health problems for the host [7].

In recent years, a growing number of studies have shown that people’s health is closely related to the micro-ecosystem, and any imbalances will alter the immune systems and metabolisms of humans, which not only affects the initiation and progression of malignant tumors, but also has a certain impact on tumor treatment response and complications associated with tumor treatment. A major part of the mechanism depends on the following three categories: (1) Integrating into host genomes. Integration may cause the mutations of oncogenes and tumor suppressor genes or alternation of genome stability, which can lead to disrupting of the balance between proliferation and apoptosis in host cells. (2) Involving the response of the immune system. Tumors may develop when the balance of the host immune system is disrupted, and an increase in malignant tumor may occur when the host immune system is suppressed. (3) Affecting the metabolic process of host food or drugs, as represented by SCFAs.

However, there are only a few publications summarizing the relationship between intestinal microbes and the initiation and progression of hematological malignancies. In this review, we primarily discuss the progression of intestinal microbes and hematological tumors, as well as the correlation of therapeutic efficacy with intestinal microbes, aiming to better understand the mechanisms involved in the development of hematological tumors and to provide potential therapeutic targets that will help improve patient survival in the future.

## 2. Leukemia

Leukemia arises from clonal proliferation of abnormal hematopoietic stem cells, leading to disruption of normal marrow function and marrow failure, with an incidence of 0.97% [8], which is characterized by uncontrolled proliferation of the malignant clone and marrow failure. Leukemia cells can be classified into acute and chronic types based on their differentiation degree and natural course. Acute leukemia (AL), derived from primitive hematopoietic stem cells and early naive cells, exhibits rapid progression and a shorter disease duration, and it can further be classified as acute lymphoblastic leukemia (ALL) and acute myeloid leukemia (AML) on the basis of cell type. Chronic leukemia (CL) is characterized by stalled differentiation of mature naive and mature cells, showing a slow progression, with a several-years natural disease course, and it includes chronic lymphocytic leukemia (CLL), chronic myeloid leukemia (CML), and rare types of leukemia, such as hairy cell leukemia and prolymphocytic leukemia [1]. The mortality rate caused by malignant tumors ranked leukemia 8th (male) and 10th (female), while it was the first-ranked cause of malignant-tumor-related mortality for children and adults under 35. There is no clear etiology for leukemia, but recent studies have revealed that intestinal microbiota play an important role in leukemia initiation, progression, and prognosis, as well as treatment effectiveness.

### 2.1. Gut Microbes and Leukemia: Clinical Relevance

At present, leukemia patients’ intestinal microbial composition has been analyzed by six research groups using 16S rRNA sequencing. It was found that leukemia patients’ intestinal microbes differ greatly from those of healthy donors, with a decrease in the diversity and variability of the flora, mainly manifested by the significant enrichment of *Proteus*, *Scleroderma*, and *Bacteroides* in leukemia patients. The increase in *Firmicutes*/*Bacteroidetes* ratio was linked with insulin resistance in clinical studies involving obese individuals with CLL. See Table 1 for details.

By 16S rRNA sequencing, Mohammed Kawari et al. compared intestinal microorganisms among six healthy donors and six CLL patients. In contrast to healthy donors, CLL patients had a diminished diversity and variability of intestinal flora, but enriched *Proteobacteria*, *Scleroderma*, and *Bacteroides*. In the clinical study of obese individuals with insulin resistance, the increased ratio of *Firmicutes*/*Bacteroidetes* in CLL patients was associated with insulin resistance [9].

Currently, the treatment strategies for leukemia mainly include chemotherapy, targeted gene therapy, stem cell transplantation and chimeric antigen receptor T cell immunotherapy (CAR-T), all of them have effects on the intestinal microbes of patients.

One of the most important treatments for leukemia is chemotherapy. Chemotherapy drugs can alter the composition of intestinal microbes in patients and reduce their diversity, whereas the intestinal microbes can predict the efficacy of chemotherapy and the occurrence of chemotherapy-related adverse reactions. Ling Ling Chua et al. explored temporal alterations of intestinal microbiota in seven children with ALL at the beginning, during, and after cessation of chemotherapy by 16S rRNA sequencing. Before chemotherapy, there were greater individual differences in intestinal flora among ALL patients compared with seven healthy donors, and *Bacteroidetes* and *Bacteroides* were enriched. The relative abundance of *Bacteroides* decreased in the intestinal flora when chemotherapy began, and ALL patients showed similar intestinal flora composition to healthy donors after chemotherapy was completed. However, although a complete remission of the disease status was noted in ALL patients, the gut microbiota β-diversity (analysis of the composition of microbial communities in different samples/groups) of ALL patients remained unique, and the abundance of a few bacteria was different from that of the control group. Therefore, the recovery of disturbed microbial communities in patients should be further explored [10]. Similar results were shown in Hana Hakim’s report, which included an analysis of high-throughput sequencing of 16S rRNA gene of fecal samples in ALL patients, though it was further found that the relative abundance of other groups such as *Clostridaceae* and *Streptococcaceae* increased after chemotherapy. Furthermore, this study also found that intestinal flora can be used as a predictor of common adverse reactions associated with chemotherapy, the increased relative abundance of *Proteobacteria* before chemotherapy predicted the development of neutropenia, while the predominance of *Enterococcus* or *Streptococcus* in the gut microbiota at any time during chemotherapy predicted a significantly increased risk of granulocyte deficiency with fever and diarrhea [11]. The decrease of microbial diversity during chemotherapy is also related to the efficacy of chemotherapy. Sangmin Lee et al. studied the diversity and composition of microbiota in newly diagnosed AML patients 14 days after chemotherapy and 30 days after chemotherapy by 16S rRNA sequencing, and discovered that the intestinal microbial α-diversity (a description of the abundance and diversity of microbial communities in a single sample) of patients decreased during induced remission chemotherapy, and the total reduction of intestinal microbial α-diversity after 14 days of chemotherapy was related to the degree of disease remission, but not related to age, European Leukemia Net (ELN) risk, or intestinal microbial diversity at initial diagnosis [12].

Continuous improvements in treatment protocols have enhanced the therapeutic effect of acute leukemia, achieving better results regarding the overall survival of patients with hematologic malignancies. Therefore, the long-term toxic effects after high intensity chemotherapy cannot be ignored. Ronay Thomas et al. studied 38 cured pediatric ALL populations and found obesity disease, cardiovascular disease, and other chronic illnesses are associated with an increased risk. In order to explore the possible mechanism, 16S rRNA sequencing was performed to examine intestinal microbe differences between ALL patients and their siblings, and identified that the abundance of *Faecalibacterium* in intestinal flora of ALL was significantly absent, which suggests that *Faecalibacterium* may be connected to the occurrence of long-term chronic diseases after chemotherapy [13].

Hematopoietic stem cell transplantation (HSCT) is an important treatment that may cure hematological malignancies. Several studies have found significant changes in the gut microbiome of patients with hematological malignancies undergoing HSCT, including significant reductions in bacterial diversity and higher expression levels of antibiotic resistance genes (ARG) after treatment. In addition, relevant studies have shown that treatment-induced alterations in the intestinal microbiome were associated with poor prognosis in patients undergoing HSCT, especially in Graft-versus-host disease (GVHD). Nevertheless, the current mechanism is unclear and deserves further exploration by researchers [14].

In short, the intestinal microbiota of leukemia patients exhibit lower diversity and variability than the healthy donors, and the microbes of *Proteobacteria*, *Firmicutes*, and *Bacteroidetes* were significantly enriched in leukemia patients. Intestinal microbes are also involved in the treatment of leukemia; chemotherapy drugs could change the composition of intestinal microbes in leukemia patients and reduce the diversity, and intestinal microbes could predict the efficacy of chemotherapy and the occurrence of chemotherapy-related adverse reactions. HSCT could also change the diversity of intestinal flora and predict the poor prognosis of leukemia patients.

**Table 1 cancers-15-02284-t001:** Relationship between intestinal microbes and leukemia.

Cancer	Detection Method	Sampling Materials	Main Findings	Study
PMP	16S, qPCR	Mouse model	In TET 2 knockout mouse model, the lack of TET 2 can cause intestinal barrier damage, lead to intestinal microbial translocation, such as translocation of *Lactobacillus reuteri*, then cause immune activation and lead to an increase in hematopoietic stem cell self-renewal ability, so that its bias to the myeloid development eventually causes pre-leukaemic myeloproliferation (PMP), promoting the initiation of leukemia.	Meisel, M. et al., 2018 [15]
ALL	16S	Human	*Bacteroides genus* is enriched in patients.	Chua, L.L. et al., 2020 [10]
ALL	16S	Human	*Enterococcaceae* abundance is related to increased risk of granulocyte deficiency with fever and diarrhea; *Streptococcidae* abundance is associated with diarrhea.	Hakim, H. et al., 2018 [11]
ALL	16S	Mouse model	The relative abundance of *Lactobacillus reuteri* is reduced in leukemic mice, and inhibits proliferation of supplement with *Lactobacillus reuteri*; *Parabacteroides goldsteinii/ASF 519* abundance is enriched in leukemic mice, damages intestinal balance, and promotes progression.	Bindels, L.B. et al., 2016 [16]
ALL	16S	Human	*Faecalibacterium* abundance shows a positive correlation with chronic diseases.	Thomas, R. et al., 2020 [13]
AL	16S	Human	Bayesian analysis of longitudinal data demonstrated larger departures of microbial communities from the pre-chemotherapy baseline during repeat therapy compared to induction. This increased ecosystem instability during repeat therapy possibly impairs colonization resistance and increases vulnerability to *Enterococcus* outgrowth.	Rashidi, A. et al., 2019 [17]
AML	16S	Human, mouse model	*Anaerostipes, Lachnospiraceae*, and *Bacteroidales S24–7* are significantly reduced in leukemic mice, which inhibits progression and maintains intestinal balance by producing SCFAs, which promote insulin secretion and inhibit glucose uptake by AML cells.	Ye, H. et al., 2018 [18]
AML	16S	Human	Negative correlation between efficacy and α-diversity.	Lee, S. et al., 2019 [12]
CLL	16S	Human	*Firmicutes* and *Bacteroidetes* show a positive correlation between ratio and insulin resistance; *Proteobacteria* is enriched in patients.	Kawari, M. et al., 2019 [9]

### 2.2. Intestinal Microbial Imbalance and the Initiation and Progression of Leukemia

#### 2.2.1. Imbalance of Intestinal Flora Activates Inflammatory Factors to Promote the Initiation of Leukemia

Previous studies have shown that TET gene mutations can promote the development of many tumors, especially tumors in the hematopoietic system [19,20,21]. TET 2 is a tet-methylcytosine dioxygenase 2, which can catalyze the conversion of 5-methylcytosine (5-mC) into 5-hydroxymethylcytosine (5-hmC). It plays a crucial role in DNA demethylation as an enzyme, which is essential for maintaining stem cell pluripotency [22]. The absence of TET 2 can cause intestinal barrier damage in a TET 2 knockout mouse model, lead to intestinal microbial translocation, such as translocation of *Lactobacillus reuteri*, then, the immune system is activated, resulting in increased stem cell self-renewal, which polarizes the development of the myeloid cells [19,20,21,23,24], that eventually trigger pre-leukaemic myeloproliferation (PMP) and promote the initiation of leukemia [19,20,22,25]. Mechanically, small-intestinal barrier dysfunction (reduced ZO-1 and upregulation of defence response genes), which occurs spontaneously or upon intestinal damage, results in bacterial translocation and to high levels of IL-6. Systemic microbial signals can bypass bacterial translocation in Tet2-deficient mice. Microbial-induced IL-6 is sensed by Tet2^−/−^ myeloid progenitor (MP) cells that overexpress IL-6Rα and are highly sensitive to IL-6 (Stat3 (pY705)). The MPs then expand upon IL-6 signals and differentiate preferentially into mature myeloid cells. This cycle results in the development of PMP. Treatment with antibiotics or neutralizing anti-IL-6 antibody can revert PMP, indicating that microbial inflammatory signals are required for PMP in the context of Tet2 deficiency [15]. This study proved that intestinal flora-dependent inflammation is necessary for the development of PMP; however, it remains to be determined whether inflammation caused by intestinal flora can result in PMP developing into leukemia.

#### 2.2.2. Short-Chain Fatty Acid Producing Bacteria Inhibit the Progression of Leukemia

Cancer-associated *cachexia* (CAC) is a multifactorial syndrome of persistent skeletal muscle mass loss, characterized by reduced intake and abnormal metabolism, leading to the breakdown of protein and a breaking of synthesis balance [26]. The role of gut microbiota in the treatment of cancer and related cachexia was reported by Laure B. Bindels et al., it was discovered that the abundance of *Lactobacillus* spp. in the intestinal flora of ALL mice decreased, while the abundance of *Enterobactereae* and *Paraacteroides goldsteinii/ASF 519* increased. ALL mice were injected with synbiotic or SCFA-producing bacteria *Lactobacillus reuteri*, and synbiotic or SCFA-producing bacteria were found to restore *Lactobacillus* abundance accompanying reduced *Enterobacteriaceae* levels, resulting in enhanced survival. The mechanism may be that synbiotic biotics or short-chain fatty acid-producing bacteria can improve intestinal barrier function (e.g. ZO-1, Muc2), increase the expression of intestinal antibacterial genes (e.g. Lasozyme, TCF4), and promote immune response (e.g. TNFa, CD3g) in ALL mice. This study suggested that nonintestinal tumors affect the gut microbial ecosystem, and intestinal homeostasis can also be preserved by specific gut microorganisms that produce short chain fatty acids [16].

It is well known that cell bioenergetics in malignant cells involve more glucose consumption than in normal somatic cells. To successfully compete for a limited amount of whole-body glucose, malignant cells need to reduce glucose availability in normal tissues such as adipose tissue and muscle, both of which consume a significant amount of glucose. This systematic rebalancing of biological processes is known as adaptive homeostatic, and similar biological processes have been reported previously, where stimulatory signals, such as aging or environmental stress, lead to an expansion or contraction of the homeostatic range. [27,28]. However, the effects of intestinal microbes and tumor cells on glucose utilization are rarely reported. Haobin Ye et al. used 16S rRNA sequencing to find that in the feces of AML mice, SCFA-producing bacterias *Lachnospiraceae* and *Bacteroidales S24-7* families were significantly reduced, and the abundance of *Anaerostipes*, which generate butyric acid, was also decreased. Additionally, there was a significant decrease in SCFAs such as butyric acid and propionic acid in the stool samples of leukemic mice. Furthermore, leukemic mice were treated with butyrate or propionate to test its functional relevance, and whether intestinal epithelial integrity was partially restored or not was recorded. Strikingly, butyrate suppressed tumor burden in bone marrow and gonadal adipose tissue, and propionate suppressed tumor burden in gonadal adipose tissue. Additionally, butyrate and propionate treatment decreased serum IGFBP1 levels, increased insulin levels, and decreased plasma glucose concentrations. The results of this study indicated that leukemia cells hijack host glucose by inducing IGFBP1 production from adipose tissue to mediate insulin sensitivity and by inducing gut dysbiosis, serotonin loss, and incretin inactivation to suppress insulin secretion. Disrupting this adaptive homeostasis attenuates leukemia progression, and that SCFAs inhibit leukemia-induced adaptive homeostasis by affecting the integrity of the intestinal epithelium and regulating IGFBP1 as well as insulin levels [18].

Briefly, the imbalance of intestinal microbiota may promote the development of leukemia by activating inflammatory factors and activating abnormal glucose metabolism. Loss of TET 2 causes gut microbial translocation that causes PMP development and may eventually lead to leukemogenesis. The absence of SCFAs producing bacteria promotes the progression of leukemia, and transplantation of such bacteria or treatment with SCFAs can significantly inhibit leukemia progression and improve survival (Figure 1). Interestingly, there is an observation that *Lactobacillus reuteri* has completely opposite functions in leukemogenesis and development, as well as completely opposite results.

### 2.3. The Treatment Prospect for Leukemia

The gut ecosystem is the largest and most complex micro-ecosystem in the human body. Homeostasis of the intestinal microenvironment plays a major role in maintaining normal physiological functions, and the imbalance of intestinal microecology has been considered to be an important factor affecting the occurrence and development of many diseases. Therefore, we may relieve disease symptoms or disease progression by the combination of drugs and interventions of the intestinal flora. By comparing the structure of intestinal flora in leukemia patients and healthy donors, studying the alterations in the structure of intestinal flora in leukemia patients will contribute to the early prevention and diagnosis of leukemia. At the same time, we can repair the unbalanced intestinal microbes by adding probiotics and adjusting the diet structure (prebiotics, SCFAs, etc.), to improve the treatment effect of leukemia and provide neoadjuvant therapy strategies in clinical care.

## 3. Lymphoma

An anaplastic disease originating in lymph nodes or lymphoid tissue, lymphoma is a heterogeneous group of neoplastic diseases. Lymphoma is a malignant tumor of the immune system that occurs when lymphocyte proliferation and differentiation are altered malignantly during the immune response process, with an incidence of 1.4% of all tumors, according to the 2018 Global Cancer Report [8], and it can occur at any age, with people 20 to 40 years old accounting for about 50% of cases. Males are more at risk of lymphoma than women, and the incidence of lymphoma increases with age, and it varies by country. At present, it is believed that the incidence of lymphoma is the result of a variety of factors, among which viral factors involved in lymphoma, such as Epstein–Barr virus (EB), human T cell lymphoma virus (HTLV), human herpes virus type VI (HHV-6), etc., which are considered to be closely related to the incidence of lymphoma. Lymphoma can occur in almost any part of the body, is characterized by painless, progressive lymphadenopathy, and local masses are the characteristic clinical manifestations, which can be accompanied by symptoms of compression in certain organs. Lymphoma is divided into two categories: Hodgkin’s lymphoma and non-Hodgkin’s lymphoma, on the basis of the histopathological features [29].

There are few studies on lymphoma and intestinal microbes, mainly focusing on the initiation of intestinal microbes and lymphoma, the composition of intestinal microbes in lymphoma patients, and intestinal microbes as the predictive index of infection after HSCT.

### 3.1. Gut Microbes and Lymphoma: Clinical Relevance

Thus far, three research centers have analyzed the diversity and composition of intestinal microbes in lymphoma patients by the 16S rRNA sequencing method. In non-Hodgkin’s lymphoma patients, the α-diversity of intestinal microbes decreased, and the abundance of gut microbes *Barnesiellaceae*, *Coriobacteriaceae*, *Faecalibacterium*, *Christensenella*, *Dehalobacterium*, *Desulfovibrio*, *Eubacterium*, and *Sutterella* was also decreased, which was related to the occurrence of bacteremia [30]. In addition, the patients with Hodgkin’s lymphoma after treatment are enriched in *Faecalibacterium* and [*Eubacterium*] *oxidoredoucens group* compared with the healthy donors, while *Streptococcus*, *Selemonas*, *Candidatus stoquefichus*, unclassified S24-7, *Faecilitale*, and *Veillonella* are significantly enriched in healthy donors [31]. Another report among lymphoma patients treated with CD19-targeted chimeric antigen receptor (CAR)-T cell therapy, a higher relative abundance of *Bacteroides eggerthii*, *Ruminococcus lactaris*, *Eubacterium* spp. *CAG 180*, *Akkermansia muciniphila*, and *Erysipelatoclostridium ramosum* increased the probability for complete remission (CR) prediction, whereas higher relative abundances of *Bacteroides stercoris* and others increased the probability for non-response prediction [32]. See Table 2 for details.

Adoptive T cell transfer therapy with chimeric antigen receptor (CAR)-T cells represents a breakthrough immunotherapy in the treatment of hematologic malignancies [33]. Preclinical studies and human studies have established that gut microbiomes play a crucial role in human health, and its effectors may have a major impact on the efficacy and toxicities of T cell-based cancer immunotherapies. In a B cell lymphoma patient cohort from five centers in Germany and the United States (Germany, *n* = 66; United States, *n* = 106; total, *n* = 172), Christoph K Stein-Thoeringer et al. demonstrated that wide-spectrum antibiotics treatment (‘high-risk antibiotics’) prior to CAR-T therapy is associated with adverse outcomes, but this effect is likely to be confounded by an increased pre-treatment tumor burden and systemic inflammation in pre-treated patients with high-risk antibiotics. In order to resolve this confounding effect and to gain insight into antibiotics-masked microbiome signals impacting CAR-T efficacy, authors focused on patients who were not exposed to antibiotics. A significant correlation was observed between pre-CAR-T infusion *Bifdobacterium longum* and microbiome-encoded peptidoglycan biosynthesis in these patients, and CAR-T treatment-associated 6-month survival or lymphoma progression. Furthermore, predictive pre-CAR-T treatment microbiome-based machine learning algorithms trained on the high-risk antibiotics non-exposed German cohort and validated by the respective US cohort robustly segregated long-term responders from non-responders. An increased relative abundance of *Bacteroides eggerthii*, *Ruminococcus lactaris*, *Eubacterium* spp. *CAG 180*, *Akkermansia muciniphila*, and *Erysipelatoclostridium ramosum* increased the probability for CR prediction was observed, whereas higher relative abundances of *Bacteroides stercoris* and others increased the probability for non-response prediction [32].

After HSCT, infection is the most common complication. Relevant studies have found that intestinal microbes can be used as a predictor of bloodstream infection (BSI) after HSCT [30]. Studies have reported that intestinal flora (defined as a single flora accounting for at least 30% of the flora) is associated with BSI in patients receiving allogeneic-HSCT (allo-HSCT) [34,35]. However, the impact of the gut microbiome on the subsequent risk of developing BSI before treatment initiation remains unclear. Emmanuel Montassier et al. analyzed the characteristics of fecal flora collected from 28 patients with non-Hodgkin’s lymphoma before treatment by 16S rRNA sequencing method to determine the microbes that predict the risk of BSI. A total of 28 patients with non-Hodgkin’s lymphoma received allo-HSCT before chemotherapy. The stool samples were sequenced, and machine learning technology was used to identify the microbial biomarkers predicting BSI. The patients with BSI showed a decline in the overall diversity of intestinal microorganisms and a decline in the abundance of groups, including *Barnesiellaceae*, *Coriobateriaceae*, *Faecalibacterium*, *Christensella*, *Dehalobacterium*, *Desulfovibrio,* and *Sutterella*. The authors developed a BSI risk index based on the pretreated fecal microbiota using only machine learning, with sensitivity of 90% and specificity of 90%. These results suggest that the gut microbiota can identify high-risk patients prior to HSCT, that intervention of the gut microbiota to prevent BSI in high-risk patients may be a clinical adjuvant strategy, and that this approach may inspire the development of diagnosis and prognosis models based on microbiota similar to other diseases [30].

Hodgkin’s lymphoma is a malignant tumor with a high cure rate of at least 80%. To explore the intestinal microbial alterations of Hodgkin’s lymphoma, Nancy Huang et al. enrolled the members of identical twins registered in the International Twin Study of the University of Southern California as the research object. In each group of twins, one was diagnosed with Hodgkin’s lymphoma and the other was not affected. A total of 24 survivors of Hodgkin’s lymphoma and their unaffected twins provided stool samples for 16S rRNA sequencing. The analysis of sequencing results showed that there was no significant difference in α-diversity and β-diversity between survivors of Hodgkin’s lymphoma and their unaffected twins. However, in each measurement, the intestinal microbial diversity score of unaffected twins was higher than that of their Hodgkin’s lymphoma twin survivors. Between survivors of Hodgkin’s lymphoma and their unaffected twins, significant differences in intestinal microbiota can be observed in the following: *Streptococcus*, *Selemonas*, *Candidatus stoquefichus*, unclassified S24-7, *Faeclitalea*, *Veillonella*, *Faecalibacterium*, and *[Eubacteria] oxidoredogenes group*. Compared with the unaffected twins, *Faecalibacterium* and *[Eubacteria] oxidoredogenes group* are more abundant in survivors, and the remaining taxa are more abundant in the unaffected twins [31].

However, research has been performed that differs from the above. Wendy Cozen et al. analyzed the intestinal microbial diversity and composition of Hodgkin’s lymphoma survivors in 13 pairs of monozygotic and monozygotic twins by 16S rRNA sequencing. It was found that fecal microbial communities of Hodgkin’s lymphoma survivors showed less α-diversity than those of unaffected twins. There is no statistically significant difference in the indicators measuring the relative abundance of bacteria. However, when the unweighted number of the unique operational taxonomic units (OTU) is considered, the difference is significant. When the analysis is limited to OTU with abundance greater than 0.1% in at least two of the twenty-three samples analyzed, the difference is weakened, and only the difference of PD integer index of diversity is significant. These results suggest that the decrease in the diversity of intestinal flora of survivors of Hodgkin’s lymphoma may be caused by disease, treatment, or special hygienic environment [36].

Similarly, Maren Schmister et al. collected stool samples from 12 patients with non-Hodgkin’s lymphoma at the time of diagnosis, prior to administration of a therapy cycle, and every three months during a 6-month follow-up period, and used flow cytometry and 16S rRNA sequencing to detect alterations in intestinal microbial diversity of patients. The authors compared the phenotypic α-diversity metrics obtained from flow cytometry analysis to their taxonomic, 16S rRNA sequencing-based counterparts. The linear mixed effect model confirmed a statistically significant positive association between taxonomic and phenotypic α-diversity, there was no alterations of phenotypic microbial diversity in patients with non-Hodgkin’s lymphoma at the time of diagnosis, however, the α-diversity gradually decreased during chemoimmunotherapy [37].

**Table 2 cancers-15-02284-t002:** Relationship between intestinal microbes and lymphoma.

Cancer	Detection Method	Sampling Materials	Main Findings	Study
NHL	16S	Human	*Barnesiellaceae*, *Coriobacteriaceae*, *Faecalibacterium*, *Christensenella*, *Dehalobacterium*, *Desulfovibrio*, and *Sutterella* negative correlation of BSI.	Montassier, E. et al., 2016 [30]
HL	16S	Human	*Streptococcus*, *Sellimonas*, *Candidatus*, *Stoquefichus*, *Veillonella*, *Faeclitalea*, and *unclassified S24-7* enriched in HD; *Faecalibacterium* and *Eubacterium oxidoreducens group* enriched in cured patients.	Huang, N. et al., 2017 [31]
NHL	Flow cytometry, 16S	Human	α-diversity decreased during therapy.	Schmiester, M. et al., 2022 [37]
NHL	Metagenomic	Human, mouse model	*Eubacterium rectum* decreased in patients, which inhibits intestinal inflammation and inactivates TLR4/MyD88 signaling to inactivate NF-kB pathway in B cells and inhibits B cell malignant.	Lu, H. et al., 2022 [38]
Lymphoma	Metagenomic	Human	A higher relative abundance of *Bacteroides eggerthii*, *Ruminococcus lactaris*, *Eubacterium* spp. *CAG 180*, *Akkermansia muciniphila*, and *Erysipelatoclostridium ramosum* increased the probability for CR prediction, whereas higher relative abundances of *Bacteroides stercoris* and others increased the probability for non-response prediction.	Christoph K Stein-Thoeringer et al., 2023 [32]
Lymphoma	16S	Mouse model	*Lactobacillus johnsonii*, which was deficient in the more cancer-prone mouse colony, was causally tested for its capacity to confer reduced genotoxicity when restored by short-term oral transfer.	Mitsuko L. Yamamoto et al., 2013 [39]

NHL: non-Hodgkin’s lymphoma; HL: Hodgkin’s lymphoma; HD: healthy donors.

### 3.2. Intestinal Microbial Imbalance and the Lymphomagenesis

Microbe-induced tumorigenesis has been confirmed in a variety of gastrointestinal solid tumors, but was rarely studied in hematological malignancies, especially lymphoma. In order to determine the role of intestinal microbiota in the initiation and progression of lymphoma, metagenomic sequencing was performed by Haiyang Lu et al. for stool samples from patients with primary gastrointestinal lymphoma and healthy donors. The unique microbial characteristics of intestinal lymphoma were identified, with a significantly reduced amount of symbiotic microorganisms, notably *Eubacterium rectum*, which produces butyrate. Furthermore, the author transferred the defective microbiota of *Eubacterium rectum* from patients with intestinal lymphoma into mice and caused inflammation and the production of tumor necrosis factor (TNF). On the contrary, treatment with *Eubacterium rectum* reduced TNF levels and lymphoma incidence rate in sensitized Eμ-Myc mice. In addition, lipopolysaccharide (LPS) in the resident microbiota of lymphoma patients and mice interacted with TNF signal and enhanced the NF-kB pathway through MyD88-dependent TLR4 signal, which also enhanced the survival and proliferation of intestinal B cells. However, in the GI tract, a *Eubacterium rectale*-deficient gut microbiota stimulated B cells through enhanced TNF release and further sensitized B cells to LPS; these extracellular substances could bind to membrane TNFR1 and TLR4 and activate NF-kB signaling in a MyD88-dependent manner, as an extrinsic mechanism. These findings reveal the mechanism of inflammation-related lymphoma and the potential clinical principle of intestinal microbiota therapy targeting [38].

The ataxia-telangiectasia (A-T) syndrome is autosomal recessive disorder with an increased likelihood of lymphoid malignancies [40]. Neoplasia occurs in approximately 30–40% of all A-T patients [41]: more than 40% of tumors are non-Hodgkin’s B cell lymphomas, about 20% acute lymphocytic leukemias, and 5% Hodgkin’s lymphomas [42]. Biallelic mutations in the ATM gene cause A-T, and there have been over 600 different ATM mutations described. Using an A-T mouse model, Mitsuko L. Yamamoto et al. discovered that the intestinal microbiota is a major contributor to disease penetrance, lifespan, molecular oxidative stress, and systemic leukocyte toxicity by comparing lymphoma incidence in several isogenic mouse colonies harboring different bacterial communities. The relative abundance of *Lactobacillus johnsonii* decresed in A-T mice by high-throughput sequencing of 16S rRNA. By restoring *Lactobacillus johnsonii* to the cancer-prone mouse colony, short-term oral transfer was causally tested for the ability to confer reduced genotoxicity. The intervention decreased systemic genotoxicity, which is associated with a reduction in basal leucocytes and cytokine-mediated inflammation, and mechanistically linked to the host cell biology of systemic genotoxicity. According to these findings, the intestinal microbiota can serve as a potential target for translational interventions in individuals at risk for B cell lymphoma, or for other diseases that are driven by genotoxicity or the molecular response to oxidative stress [39].

In conclusion, the abundance of gut microbiota can be used as a predictor of BSI in lymphoma patients, and the deletion of butyric acid-producing bacterium *Eubacterium rectum* may cause lymphoma, as well as the deficiency of *Lactobacillus johnsonii* may cause lymphoma in A-T mice (Figure 2). However, further research on gut microbes and lymphoma needs to be performed.

### 3.3. The Treatment Prospect for Lymphoma

There are few studies on intestinal microbes in lymphoma, whether their microbial composition at initial diagnosis or after treatment. Lymphoma patients usually lack *Eubacterium rectum* in the intestine, leading to inflammation and lymphoma development. Lymphoma initiation can be reduced by supplementing *Eubacterium rectum*, whose absence can be used to indicate lymphomagenesis risks. In addition, in an A-T mouse model, the deficiency of *Lactobacillus johnsonii* may lead to the initiation of lymphoma. As a result of lymphoma disease characteristics, the immunity of most lymphoma patients is suppressed, resulting in infection and death in severe cases; therefore, clinical lymphoma patients will use prophylactic antibiotics, and the blind use of antibiotics may make the bad intestinal microbes worse, thereby affecting the survival of patients. The research results of Emmanuel Montassier et al. have good reference and guiding significance for solving this problem and for newly diagnosed patients to assess the risk of infection and guide them regarding the possible use prophylactic antibiotics. However, this method has defects. According to a prospective clinical study, a higher relative abundance of *Bacteroides eggerthii*, *Ruminococcus lactaris*, *Eubacterium* spp. *CAG 180*, *Akkermansia muciniphila*, and *Erysipelatoclostridium ramosum* increased the probability for CR prediction, whereas higher relative abundances of *Bacteroides stercoris* and others increased the probability for non-response prediction. The composition of intestinal microorganisms in lymphoma patients is not clear, and whether it is universal or whether the use of antibiotics will make the imbalance of intestinal microorganisms in patients more serious are urgent problems to be solved.

## 4. Multiple Myeloma

Multiple myeloma (MM) is a terminally differentiated malignant clonal disease of plasma cells with an incidence of 0.41% of all human tumors, manifested by bone marrow clonal plasma cell infiltration and the presence of monoclonal M protein in peripheral blood and (or) urine [8]. The clinical manifestations are osteolytic damage, renal function damage, hypercalcemia, repeated infection, anemia, etc. The risk factors include aging and environmental factors, which tend to occur in the elderly. The peak age of incidence is 50 to 60 years old, with the increase of the elderly, there is a trend of increasing year by year. [43]. The latest studies suggest potential links between gut microbiota and various diseases such as MM.

### 4.1. Gut Microbes and Multiple Myeloma: Clinical Relevance

Two research teams analyzed the gut microbial diversity and composition of MM patients through metagenome sequencing and 16S rRNA sequencing. It was found that the intestinal microbial diversity and microbial abundance of newly diagnosed MM patients were significantly different from those of healthy donors. The intestinal microbes of MM patients had higher α-diversity at the genus level and species level, *Anaerostipes hadrus*, *Clostridium saccharobutylicum*, and *Clostridium butyrate* were significantly enriched in healthy donors, and *Citrobacter freundii*, *Klebsiella variicola*, *Klebsiella pneumoniae*, and others were significantly enriched in MM patients. Moreover, the *Enterobacteriaceae* in *Proteobacteria* and *Streptococcaceae* in *Firmicutes* were positively correlated with the ISS stage of MM patients [44], and intestinal *Klebsiella pneumonia* is significantly enriched in MM with pneumonia [45]. *Eubacterium hallii*, which belongs to the *Firmicutes*, together with *Eubacteriaceae* and *Eubacterium*, were related to the negative status of minimal residual disease (MRD) in MM patients after initial treatment [46]. See Table 3 for details.

In the study of Xingxing Jian et al., the author analyzed the diversity of intestinal microbes in 18 healthy donors and 19 newly diagnosed MM patients by using the method of metagenomic sequencing, and found that the newly diagnosed MM patients were significantly enriched in *Raoultella ornithinolytica*, *Citrobacter freundii*, *Enterobacter cloacae*, *Klebsiella aerogenes*, *Klebsiella variicola*, *Klebsiella pneumoniae*, *Streptococcus salivarius*, *Streptococcus oralis*, *Streptococcus gordonii*, *Streptococcus mitis*, and *Streptococcus pneumoniae* compared with healthy donors, while the healthy donors were significantly enriched in *Anaerostipes hadrus*, *Clostridium saccharobutylicum,* and *Clostridium butyrate*. In addition, *Enterobecteriaceae* and *Streptococcaceae* were positively correlated with the ISS stage of MM patients, indicating that their abundance was positively correlated with the progression of MM [35]. Following this, the team conducted further research, and found that intestinal *Klebsiella pneumonia* is significantly enriched in MM with pneumonia, and intestinal *Klebsiella pneumonia* indirectly contributes to pneumonia in MM by synthesizing glutamine [45]. MM patients achieve MRD negativity after prior treatment and have a better prognosis compared with MRD positive patients. Recently, the connection between several specialized symbiotic microbes and the MRD of plasma cell diseases have been established. MRD negative patients had a higher abundance of butyrate-producing bacteria *Eubacterium hallii* compared with MRD-positive patients by analyzing the difference of intestinal microbes in 34 patients with MM after treatment. Alternatively, the authors identified *Faecalibacterium prausnitzii* as another microbe that may be associated with MRD positive status after initial treatment with MM. This suggests that there is a potential association between microbial composition and MM recurrence, which may be used as a clinical indicator of MRD in MM patients, or even as a predictor of recurrence when clinical validation has been completed [46].

In brief, *Enterobacteriaceae* and *Streptococcaceae* abundance can be used as indicators to predict MM progression and *Eubacterium hallii* as indicators to judge MRD-negative patients.

### 4.2. Intestinal Microbe Imbalance and the Progression of Multiple Myeloma

#### 4.2.1. Alterations of Gut Microbiome Accelerate Multiple Myeloma Progression by Increasing the Relative Abundances of Nitrogen-Recycling Bacteria

Alterations in gut microbiota are closely related to human health, and have an important relationship with the occurrence and development of various diseases. Despite extensive investigator studies of the risk factors for MM, little is known about the role of the gut microbiota and its metabolic functions in the pathogenesis of MM.

It was found that the intestinal flora of MM patients were significantly different compared with healthy donors, especially nitrogen-recycling bacteria were significantly enriched in MM patients, such as *Klebsiella* and *Streptococcus*. Meanwhile, microbes with differential abundance were also found to be significantly associated with differential metabolites of host serum, suggesting an association between gut flora and host metabolism. Subsequently, fecal microbiota transplantation (FMT) was performed in 5TGM1 MM mice and it was found that the tumor burden of mice transplanted with fecal bacteria from MM patients increased significantly, and higher concentrations of glutamine were detected in the bone marrow, serum, and cecal contents of mice. Further single bacterial transplantation (wild-type *Klebsiella pneumoniae* and glnA-mutation *Klebsiella pneumoniae*) showed that the tumor burden of 5TGM1 mice transplanted with wild-type *Klebsiella pneumoniae* was higher, which implied that glutamine produced by *Klebsiella pneumoniae* leads to increased tumor burden, and the fact that MM progression was relieved in 5TGM1 mice fed a glutamine-deficient diet also support it. Therefore, this study is the first to propose a new mechanism of promoting MM progression by the imbalance of bone marrow/peripheral blood metabolite-intestinal flora in MM: the abnormality of amino acid in the bone marrow microenvironment of the MM and a large number of light chain proteins cause renal tubular damage, resulting in the accumulation of blood urea, and the proliferation of nitrogen-recycling bacteria. On the contrary, nitrogen-recycling bacteria cause urea degradation, glutamine synthesis and absorption by the MM cells, thus accelerating the tumor progression. This study revealed a new mechanism of intestinal flora alteration in accelerating MM progression, suggesting that the intervention of the intestinal flora in MM patients may help improve traditional tumor therapy. In this article, the authors also used the *Clostridium butyrate* enriched in HC to colonize the intestine of 5TGM1 mice, and the experimental results showed that *Clostridium butyrate* can significantly inhibit MM progression, but the mechanism was not thoroughly studied by the authors [44].

#### 4.2.2. The Imbalance of Intestinal Flora Activates Th17 Cells to Promote the Process of Multiple Myeloma

Arianna Calcinotto et al. reported that in the mouse model Vk * MYC, *Prevotella hepatica* promoted the differentiation of Th17 cells in the intestine and migrated to the bone marrow to secrete IL-17, thus promoting MM progression, suggesting that commensal bacteria release a paracrine signaling network between innate and adaptive immunity to accelerate MM progression. It describes that B cells randomly acquire the characteristics of MM and migrate to the bone marrow after the activation of MYC, dependent on the germinal center. Meanwhile, a favorable cytokine environment induces Th17 differentiation and eosinophils (Eos) activation in the bone marrow niche, thus establishing a positive feedback loop of self-amplification and maintaining the progress of MM, and the specific intestinal microbes are conducive to the differentiation of Th17 cells, which migrate to bone marrow and further promote the eos-Th17-MM cell network [47].

In conclusion, gut microbes can promote MM progression through mechanisms such as metabolic reprogramming and immune activation, and gut microbial abundance may provide new insights into MM progression (Figure 3).

### 4.3. The Treatment Prospect for Multiple Myeloma

Currently, there are few reports on the relationship between MM and intestinal microbes, which may be related to its low incidence and the difficulty in collecting patient specimens. The accumulation of urea in MM patients led to the increase of nitrogen-recycling bacteria, and the amino acids synthesized by nitrogen-recycling bacteria were used by MM cells to promote the MM progression, and MM development was been inhibited when targeted the nitrogen-recycling bacteria *Klebsiella pneumoniae* glutamine synthetase gene. The MM progression effects of gut microbes and metabolites were examined for the first time, and the gut microbial composition and metabolic situation of MM patients were investigated, which has constructive significance and provides a neoadjuvant for the diagnosis and treatment of MM. Arianna Calcinotto et al. reported that in Vk * MYC mice, *Prevotella hepatica* promoted the progress of MM by promoting the differentiation of Th17 cells in the intestine and migrating to the bone marrow, which has guiding significance for clinical treatment of MM. Matthew J. Pianko et al. suggested that *Eubacterium hallii* acts as an indicator for MRD negativity. The above research results suggest that MM may be treated with drugs targeted at a certain microbe or a combination of dietary management and chemotherapy drugs, and the curative effect and prognosis of MM can be predicted according to the abundance of a certain microbe, so as to improve the accuracy of diagnosis and reduce the medical expenses of patients. Therefore, it may be a worthwhile research project for treating MM with intestinal flora as the target.

## 5. Conclusions

The intestinal microbiome, as the second genome of human body, plays an important role in maintaining normal physiological functions in the human body, and hematological tumor progression and initiation are influenced by the balance of intestinal microbes. Therefore, based on published research, we may decrease the initiation of hematological malignancies and relieve the progression of hematological malignancies by regulating intestinal flora. A significant difference in intestinal microbial diversity exists between patients with hematological malignancies and healthy donors, and the SCFAs- producing microbes are reduced or absent in most patients of hematological malignancies. The absence of probiotics and prebiotics may cause the excessive proliferation of pathogenic bacteria and intestinal microbial disorders, inflammation, and even hematological tumor development. Significant epidemiological data have established that subjects with decreased SCFA levels have an increased risk of inflammatory disorders and cancer [48]. It has been shown that SCFAs in the host gut and other parts can inhibit the growth and migration of cells, suppress histone deacetylases (HDAC), and induce apoptosis, thus reducing carcinogenesis extensively and providing future assistive strategies for the prevention and treatment of tumors [49]. In leukemia, several studies reported that prebiotics or probiotics can restore intestinal barrier function, enhance the expression of intestinal antimicrobial proteins, and promote immune response [16], further inhibit proliferation of leukemia cells and improve survival with increased insulin levels, and decreased plasma glucose concentrations [18]. However, in certain situations, such as when the TET2 gene is deleted, translocation of probiotic *Lactobacillus reuteri* may lead to PMP, which may induce leukogenesis [15]. In lymphoma, deficiency of probiotics *Eubacterium recale* may increase the incidence rate of lymphoma, while supplementation with it may reduce lymphomagenesis [38]. The supplementation of probiotic *Lactobacillus johnsonii* may reduce the incidence of lymphoma in cases of A-T [39]. In multiple myeloma, sustained minimal residual disease negativity in multiple myeloma is associated with stool butyrate [50], and there is evidence that butyrate can induce apoptosis in MM cells [51]. Additionally, probiotic *Clostridium butyrate* can inhibit MM progression when supplemented [44]. Therefore, functional foods such as prebiotics (e.g., SCFAs) and probiotics (e.g., *Clostridium butyrate*) have beneficial influences in controlling and preventing cancer in animal and preclinical models.

Probiotics, such as *Lactobacillus*, may be effective in preventing infections in patients with suppressed immune systems. A phase I trial is studying the side effects and how well giving enteral nutrition, including *Lactobacillus*, works in preventing infections in patients undergoing donor stem cell transplant for hematologic cancer or myelodysplastic syndrome (NCT00946283). Another trial is investigating the effect of fermented milk supplementation on symptoms of disease and treatment in patients with multiple myeloma. Patients with multiple myeloma may experience symptoms related to the disease and/or treatment that affect the quality of life. Supplementing one’s usual diet with a probiotic fermented milk product called kefir may contribute to reducing disease and treatment-related side effects through changing the intestinal bacteria community structure and related metabolism (NCT04530812). The clinical trial about butyrate, which belongs to prebiotics, investigates whether the tolerance has been built for up to three 21-day cycles with Arginine Butyrate and ganciclovir/valganciclovir, or results in responses in patients with EBV (+) lymphoid malignancies (NCT00917826). This is an interesting approach since, in general, cancer survivors experience more rapid declines in health-related quality of life, including physical and psychological comorbidities [52]. Therefore, prebiotics and probiotics are known to impact the composition and functional capacity of the microbiota with potential health benefits, which might be an adjuvant tool to rebalance the microbiota composition and improve inflammation and immune function, and cancer therapy will go beyond raising hope in the near future with a better understanding of cancer initiation and development.

Gut microbes are also closely related to the initiation and progression of hematological malignancies. Direct and indirect mechanisms influence the initiation and progression of hematological malignancies by intestinal microbes. In direct mechanisms, gut microbes enter the bloodstream and activate IL-6 signals of the host to promote hematological tumor progression [15]. In indirect mechanisms, metabolic reprogramming and immune activation are involved. The metabolite productions generated by intestinal microbes contribute to the alteration of plasma glucose concentrations and amino acid metabolism, resulting in tumor growth [18,44,53,54]. The activation of the immune system is usually caused by direct contact between intestinal immune cells and microbial metabolites, which activates TLR4/MyD88/NF-kB signaling in B cells [38] and the immune response, and promotes the progression of hematological malignancies. Specific intestinal microbes are conducive to the differentiation of Th17 cells, which migrate to bone marrow and further promote the eos-Th17-tumor cell positive-feedback loop [47] and sustain the progression of hematological malignancies. In this way, we can formulate appropriate prevention or treatment measures based on a better understanding of the hematological cancer-promoting mechanism. However, there is still a lack of research on intestinal microbes of hematological malignancies, compared with research on solid tumors. Numerous factors play a role in the initiation of hematological tumors, including: genetic, environmental, dietary, and microbial factors. It is nevertheless clear that genetic, environmental, and dietary factors can also affect intestinal microbes, resulting in greater heterogeneity of intestinal microbes among tumor patients. Whether the results obtained by researchers are universal is an urgent problem to be solved. As for intestinal microbes and tumorigenesis, it is still unknown which is the cause and which is the result.

In general, intestinal microbes and the initiation and progression of hematological tumors are inseparable, but the relationship between the intestinal microbes and hematological tumors cannot be proven without more evidence.

## Figures and Tables

**Figure 1 cancers-15-02284-f001:**
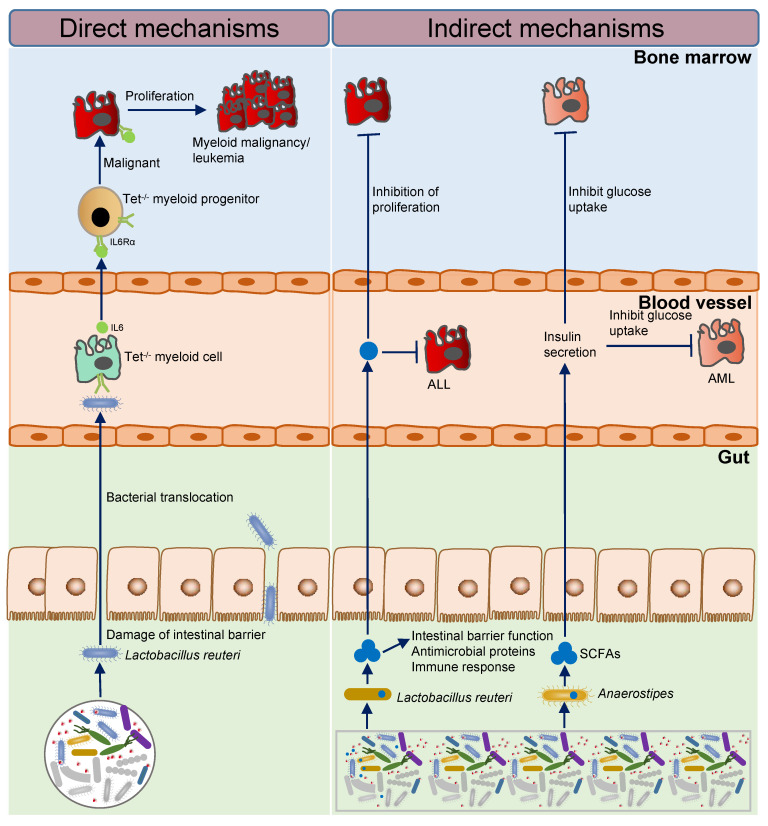
The mechanism of intestinal microbes affecting the initiation and progression of leukemia.

**Figure 2 cancers-15-02284-f002:**
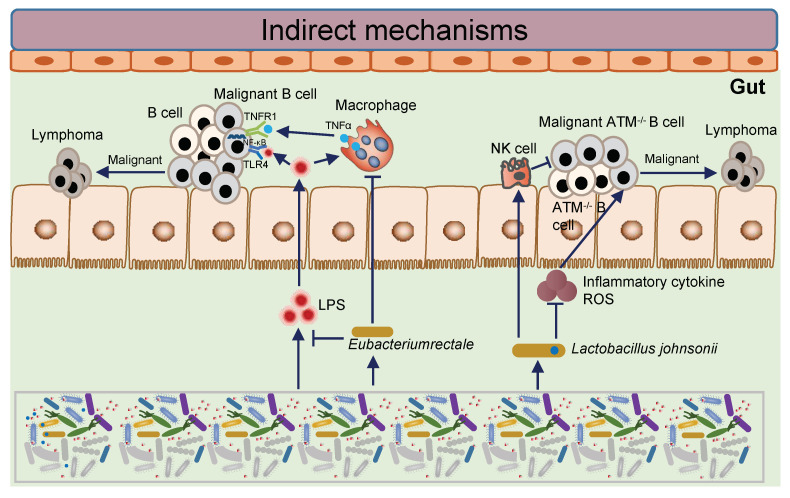
The mechanism of intestinal microbes affecting the initiation and progression of lymphoma.

**Figure 3 cancers-15-02284-f003:**
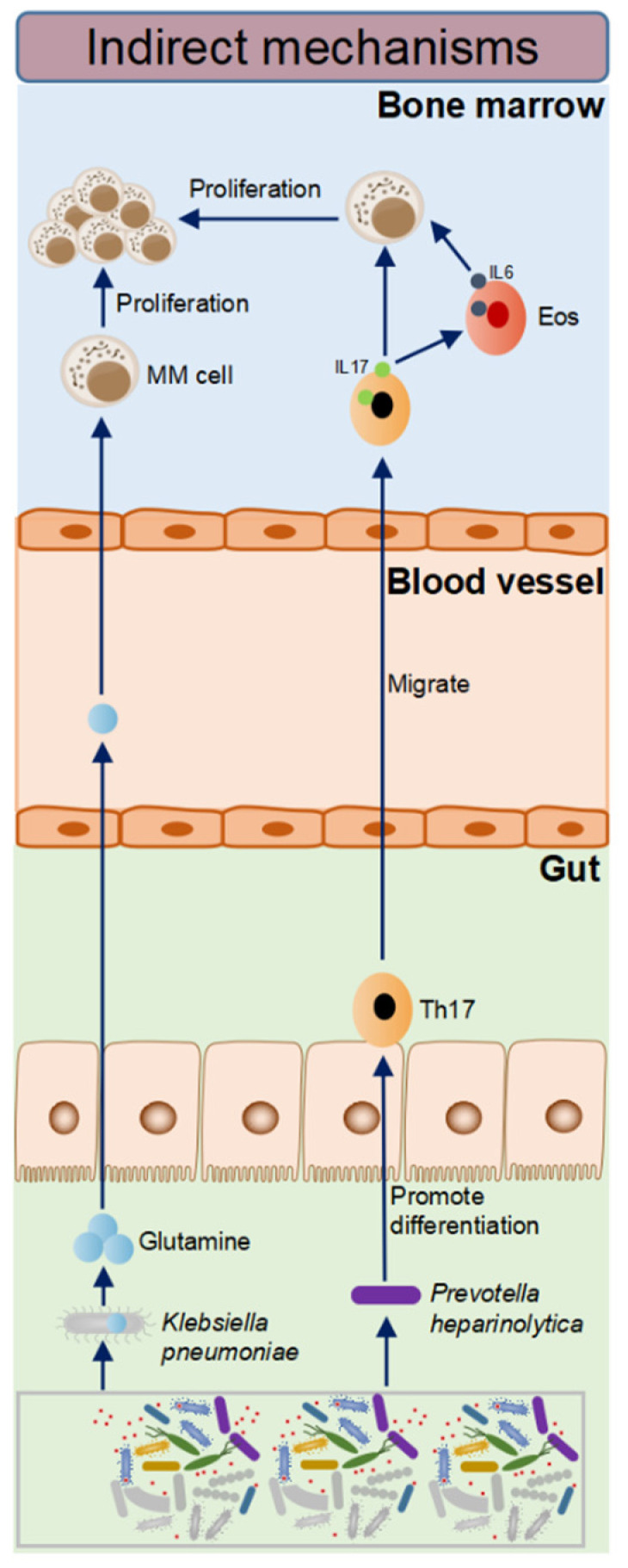
The mechanism of intestinal microbes affecting the progression of multiple myeloma.

**Table 3 cancers-15-02284-t003:** Relationship between intestinal microbes and MM.

Cancer	Detection Method	Sampling Materials	Main Findings	Study
MM	16S	Mouse model	*Prevotella heparinolytica* promotes Th17 differentiation, and Th17 migrates to the BM niche, and further contributes to the eosinophil-Th17-MM cells network.	Calcinotto, A. et al., 2018 [47]
	16S	Human	*Eubacterium hallii* is enriched in MRD negative patients.	Pianko, M.J. et al., 2019 [46]
	Metagenomic, qPCR	Human, mouse model	*Clostridium butyricum,* and *Anaerostipes hadrus* is enriched in HD, and *Clostridium butyricum* inhibits MM progression; *Klebsiella*, *Streptococcus*, etc. shows positive correlation of ISS stage; Glutamine synthesized by *Klebsiella pneumoniae*, and which belongs to nitrogen-recycling bacteria, and glutamine promotes proliferation of MM cells.	Jian, X. et al., 2020 [44]

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
