# Peer review of "Intestinal Microbes and Hematological Malignancies"

_cancers, 2023, doi:10.3390/cancers15082284_

Round 1

Reviewer 1 Report

This review article by Yinghong Zhu et al. titled "Intestinal microbes and hematological malignancies" is an informative study that focuses on the advancement of intestinal microorganisms and hematological malignancies, as well as the relationship between therapeutic effectiveness and intestinal microorganisms. The manuscript is well-structured and the objectives of the article are clearly stated in view of the subject. However, there are several issues in this study:

·         A paragraph to discuss the intentional microbes of healthy donors is missing. It is suggested that the Authors add a part to their manuscript to describe in detail the micro-ecosystem in the human body and the role that homeostasis of the intestinal microenvironment plays in maintaining normal physiological functions.

·         It is recommended that the Authors include information regarding functional foods such as prebiotics and probiotics and their beneficial role in controlling the intestinal microbe in their manuscript. Furthermore, it is strongly suggested that the Authors associate probiotics and prebiotics with the progression of hematological malignancies.

·         It is proposed that the Authors relate the mechanism of intestinal flora alteration to the acceleration of the progression of hematological malignancies and provide the molecular pathways involved.

·         Authors are encouraged to include more comprehensive Figures in their manuscript to better understand the interaction between intestinal immune cells and microbial metabolites. Additionally, the addition of a figure in the Conclusion section is not usual, so Figure 1 could be cited elsewhere in the manuscript.

·         Additional commentary could be added to the Conclusions, regarding the importance of the findings and their potential utilization in standard clinical practice. Lastly, it is recommended that the authors include future research perspectives of their study as well.

·         The literature could be more up-to-date; more relevant and recent literature needs to be added.

·         Proofreading of the manuscript is needed since there are several syntax and grammar errors throughout the text.

Author Response

Response to Reviewer 1 Comments

Point 1:

A paragraph to discuss the intentional microbes of healthy donors is missing. It is suggested that the Authors add a part to their manuscript to describe in detail the micro-ecosystem in the human body and the role that homeostasis of the intestinal microenvironment plays in maintaining normal physiological functions.

Response 1:

We appreciate your valuable comments. A detailed definition was inserted in the line 48-63 marked in red, which was also presented as follows.

As the essential micro-ecosystem in the body, gut microbes maintain a long-term mutualistic relationship with the host through fermenting dietary fiber, defending against pathogens, and synthesizing short-chain fatty acids (SCFAs) or beneficial metabolites. Moreover, the host's immune system's maturation and function are also susceptible to the homeostasis of gut microbes. The role SCFAs play in intestinal homeostasis has been confirmed, and it is further correlated with tissues and organs beyond intestinal through blood circulation, SCFAs not only signal through binding to cognate G-protein-coupled receptors on endocrine and immune cells in the body but also induce epigenetic changes in the genome through effects on the activity of histone acetylase and histone deacetylase enzymes. The interaction between the mammalian intestinal and its resident gut microbiota is delicate, the gut establish tolerance with commensal microbes, maintain control over pathobionts, and keep the functions of intestinal epithelial barrier by preventing microbial overgrowth. In turn, microbiota in the gut modulate and regulate the immune system in order to promote tolerance within the host. Any disruption to this exquisite equilibrium might cause health problems for the host.

Point 2: 

It is recommended that the Authors include information regarding functional foods such as prebiotics and probiotics and their beneficial role in controlling the intestinal microbe in their manuscript. Furthermore, it is strongly suggested that the Authors associate probiotics and prebiotics with the progression of hematological malignancies.

Response 2: 

Thank you for pointing it out. A detailed definition was inserted in the line 603-625 marked in red, which was also presented as follows.

The absence of probiotics and prebiotics may cause the excessive proliferation of pathogenic bacteria and intestinal microbial disorders, inflammation, and even hematological tumor development. Significant epidemiological data have established that subjects with decreased SCFA levels have an increased risk of inflammatory disorders and cancer. It has been shown that SCFAs in the host gut and other parts can inhibit the growth and migration of cells, suppress histone deacetylases (HDAC), and induce apoptosis, thus reducing carcinogenesis extensively and providing future assistive strategies for the prevention and treatment of tumors. In leukemia, several studies reported that prebiotics or probiotics can restore intestinal barrier function, enhance the expression of intestinal antimicrobial proteins, promote immune response, increased insulin levels, and decreased plasma glucose concentrations, and further inhibit proliferation of leukemia cells and prolong survival. However, in certain situations, such as when the TET2 gene is deleted, translocation of probiotic Lactobacillus reuteri may lead to PMP, which may induce leukogenesis. In lymphoma, deficiency of probiotics Eubacterium recale may increase incidence rate of lymphoma, while supplementation with it may reduce lymphomagenesis. And the supplementation of probiotic Lactobacillus johnsonii may reduce the incidence of lymphoma in cases of ataxia-telangiectasia (A-T). In multiple myeloma, sustained minimal residual disease negativity in multiple myeloma is associated with stool butyrate, and there is evidence that butyrate can induce apoptosis in MM cells. Additionally, probiotic Clostridium butyrate can inhibit the MM progression when supplemented. Therefore, functional foods like prebiotics (e. g. SCFAs) and probiotics (e. g. Clostridium butyrate) have beneficial influences in controlling and preventing cancer in animal and preclinical models.

Point 3:

It is proposed that the Authors relate the mechanism of intestinal flora alteration to the acceleration of the progression of hematological malignancies and provide the molecular pathways involved.

Response 3:

Thank you for pointing it out. A detailed definition was inserted in the line 646-659 marked in red, which was also presented as follows.

Gut microbes are also closely related to the initiation and progression of hematological malignancies. Direct and indirect mechanisms influence initiation and progression of hematological malignancies by intestinal microbes. In direct mechanisms, gut microbes enter the bloodstream and activate IL-6 signals of the host to promote hematological tumor progression. In indirect mechanisms, metabolic reprogramming and immune activation are involved. The metabolite productions generated by intestinal microbes contribute to alteration of plasma glucose concentrations and amino acid metabolism resulting in tumor growth. The activation of the immune system is usually caused by direct contact between intestinal immune cells and microbial metabolites, which activates TLR4/MyD88/NF-kB signaling in B cells, and the immune response and promotes the progression of hematological malignancies. Specific intestinal microbes are conducive to the differentiation of Th17 cells, which migrate to bone marrow and further promote the eos-Th17-tumor cell positive-feedback loop, and sustain hematological malignancies progression.

Point 4:

Authors are encouraged to include more comprehensive Figures in their manuscript to better understand the interaction between intestinal immune cells and microbial metabolites. Additionally, the addition of a figure in the Conclusion section is not usual, so Figure 1 could be cited elsewhere in the manuscript.

Response 4:

Thank you for this comment. We have already divided the image into three parts by adjusting the cite position, which indeed made our review more clear. A detailed definition was inserted in the line 255-258, line 436-439, and line 569-572 marked in red.

Point 5:

Additional commentary could be added to the Conclusions, regarding the importance of the findings and their potential utilization in standard clinical practice. Lastly, it is recommended that the authors include future research perspectives of their study as well.

Response 5:

Thank you for this comment. A detailed definition was inserted in the line 626-645 marked in red, which was also presented as follows.

Probiotics, such as Lactobacillus, may be effective in preventing infections in patients with suppressed immune systems. A phase I trial is studying the side effects and how well giving enteral nutrition, including Lactobacillus, works in preventing infections in patients undergoing donor stem cell transplant for hematologic cancer or myelodysplastic syndrome (NCT00946283). Another one investigates the effect of fermented milk supplementation on symptoms of disease and treatment in patients with multiple myeloma. Patients with multiple myeloma may experience symptoms related to the disease and/or treatment that affect the quality of life. Supplementing usual diet with a probiotic fermented milk product called kefir may contribute to reducing disease and treatment-related side effects through changing the intestinal bacteria community structure and related metabolism (NCT04530812). The clinical trial about butyrate which belongs to prebiotics, investigates whether the tolerance has been built for up to three 21-day cycles with Arginine Butyrate and ganciclovir/valganciclovir, or results in responses in patients with EBV (+) lymphoid malignancies (NCT00917826). This is an interesting approach since, in general, cancer survivors experience more rapid declines in health-related quality of life, including physical and psychological comorbidities. Therefore, prebiotics and probiotics are known to impact the composition and functional capacity of the microbiota with potential health benefits, which might be an adjuvant tool to rebalance the microbiota composition and improve inflammation and immune function, and cancer therapy will go beyond raising hope in the near future with the better understanding of cancer initiation and development.

Point 6:

The literature could be more up-to-date; more relevant and recent literature needs to be added.

Response 6:

Thank you for this comment. Adding more literature made our review more novel and plentiful. References were also presented as follows.

  1. Lavelle, A.; Sokol, H. Gut microbiota-derived metabolites as key actors in inflammatory bowel disease. Nat Rev Gastroenterol Hepatol 2020, 17, 223-237, doi:10.1038/s41575-019-0258-z.
  2. van der Hee, B.; Wells, J.M. Microbial Regulation of Host Physiology by Short-chain Fatty Acids. Trends Microbiol 2021, 29, 700-712, doi:10.1016/j.tim.2021.02.001.
  3. Amoroso, C.; Perillo, F.; Strati, F.; Fantini, M.C.; Caprioli, F.; Facciotti, F. The Role of Gut Microbiota Biomodulators on Mucosal Immunity and Intestinal Inflammation. Cells 2020, 9, doi:10.3390/cells9051234.
  4. Stein-Thoeringer, C.K.; Saini, N.Y.; Zamir, E.; Blumenberg, V.; Schubert, M.L.; Mor, U.; Fante, M.A.; Schmidt, S.; Hayase, E.; Hayase, T.; et al. A non-antibiotic-disrupted gut microbiome is associated with clinical responses to CD19-CAR-T cell cancer immunotherapy. Nat Med 2023, doi:10.1038/s41591-023-02234-6.
  5. Yamamoto, M.L.; Maier, I.; Dang, A.T.; Berry, D.; Liu, J.; Ruegger, P.M.; Yang, J.I.; Soto, P.A.; Presley, L.L.; Reliene, R.; et al. Intestinal bacteria modify lymphoma incidence and latency by affecting systemic inflammatory state, oxidative stress, and leukocyte genotoxicity. Cancer Res 2013, 73, 4222-4232, doi:10.1158/0008-5472.CAN-13-0022.
  6. Peterson, R.D.; Funkhouser, J.D.; Tuck-Muller, C.M.; Gatti, R.A. Cancer susceptibility in ataxia-telangiectasia. Leukemia 1992, 6 Suppl 1, 8-13.
  7. Ben Arush, M.W. Treatment of lymphoid malignancies in patients with ataxia-telangiectasia. Med Pediatr Oncol 1999, 32, 479-480, doi:10.1002/(sici)1096-911x(199906)32:6<479::aid-mpo25>3.0.co;2-h.
  8. Wang, Y.; Yang, Q.; Zhu, Y.; Jian, X.; Guo, J.; Zhang, J.; Kuang, C.; Feng, X.; An, G.; Qiu, L.; et al. Intestinal Klebsiella pneumoniae Contributes to Pneumonia by Synthesizing Glutamine in Multiple Myeloma. Cancers (Basel) 2022, 14, doi:10.3390/cancers14174188.
  9. Alvarez-Mercado, A.I.; Del Valle Cano, A.; Fernandez, M.F.; Fontana, L. Gut Microbiota and Breast Cancer: The Dual Role of Microbes. Cancers (Basel) 2023, 15, doi:10.3390/cancers15020443.
  10. Xia, J.; Zhang, J.; Wu, X.; Du, W.; Zhu, Y.; Liu, X.; Liu, Z.; Meng, B.; Guo, J.; Yang, Q.; et al. Blocking glycine utilization inhibits multiple myeloma progression by disrupting glutathione balance. Nat Commun 2022, 13, 4007, doi:10.1038/s41467-022-31248-w.
  11. Wu, X.; Xia, J.; Zhang, J.; Zhu, Y.; Wu, Y.; Guo, J.; Chen, S.; Lei, Q.; Meng, B.; Kuang, C.; et al. Phosphoglycerate dehydrogenase promotes proliferation and bortezomib resistance through increasing reduced glutathione synthesis in multiple myeloma. Br J Haematol 2020, 190, 52-66, doi:10.1111/bjh.16503.

Point 7:

Proofreading of the manuscript is needed since there are several syntax and grammar errors throughout the text.

Response 7:

Thank you for this comment, we have corrected the mistakes in the line 21-24, line 40-41, line 69-73, line 75, line 84-92, line 103-105, line 117, line 120, line 124-128, line 133-135, line 151-153, line 155-160, line 167-168, line 170-173, line 182, line 186-192, line 208, line 211-218, line 223-224, line 226-230, line 232-237, line 239, line 248, line 261, line 265-266, line 271, line 275-282, line 284-285, line 288, line 294-296, line 299-301, line 339-341, line 345, line 347-348, line 357-359, line 361-373, line 376-378, line 397-401, line 403-404, line 408-411, line 433-435, line 443-449, line 455-459, line 466, line 468-473, line 477, line 480-487, line 491-496, line 499-501, line 504-508, line 509-512, line 526-527, line 531-532, line 535-539, line 548-549, line 560-563, line 568-569, line 575, line 579-588, line 592, line 596, line 599-600, and line 670-671 marked in red.

Reviewer 2 Report

The Authors present a paper: " Intestinal microbes and hematological malignancies" interesting from a clinical point of view but very weak for an application at the present.

1. The correlation between intestinal microbes and hematological malignancies onset is based on studies in mice and very few data are reported in subjects affected by Leukemia or Lymphoma or Myeloma  to make convincing and scientifically accepted this correlation.

2. The study found that intestinal flora can be used as a predictor of common adverse reactions associated with chemotherapy: it is not a new report. Also that the decrease of microbial during chemotherapy is related to the efficacy of chemotherapy.

3. The Authors mention (pag.5 and 7) "previous studies" and "relevant studies" without a correlation in the References: too vague

4. The message reported in the paper is quite dangerous for a clinician because the study is interesting but the Conclusions not easy to be accepted.

5. Functional foods like prebiotics and probiotics have beneficial influences in controlling and preventing cancer in preclinical and animal models:in which way? What practical suggestion not too generic should be addressed to clinicians?

6. Tables and figure are not easy for reading

Author Response

Response to Reviewer 2 Comments

Point 1:

The correlation between intestinal microbes and hematological malignancies onset is based on studies in mice and very few data are reported in subjects affected by Leukemia or Lymphoma or Myeloma  to make convincing and scientifically accepted this correlation.

Response 1:

Thank you for this comment. There is a lack of research on the relationship between intestinal microbes and Leukemia, Lymphoma or Myeloma at present. However, intestinal microbes and hematological malignancies have been reported in several studies, particularly alters in intestinal microbes in clinical samples. In leukemia, Haobin Ye et al. reported that in the chronic phase of disease (MDS) patients, there is already an insulin-resistant phenotype. The consumption of systemic glucose is greatly increased in acute phase (AML) patients, thus the peripheral glucose levels are reduced even with reduced insulin levels. These results were consistent with their observations in mouse models. The authors found that SCFAs producing bacteria and SCFAs significantly reduced in AML mice, and treated leukemic mice with butyrate or propionate, and a partial rescue of intestinal epithelial integrity was observed. Strikingly, butyrate suppressed tumor burden in bone marrow and gonadal adipose tissue, and propionate suppressed tumor burden in gonadal adipose tissue. In addition, butyrate and propionate treatment decreased serum IGFBP1 levels, increased insulin levels, and decreased plasma glucose concentrations. This study demonstrates that leukemia cells hijack host glucose by inducing IGFBP1 production from adipose tissue to mediate insulin sensitivity and by inducing gut dysbiosis, serotonin loss, and incretin inactivation to suppress insulin secretion. Disrupting this adaptive homeostasis attenuates leukemia progression; In lymphoma, Haiyang Lu et al. reported that Eubacterium rectum and butyrate significantly reduced in lymphoma patients. Furthermore, the author transferred the defective microbiota of Eubacterium rectum from patients with intestinal lymphoma into mice to cause inflammation and the production of tumor necrosis factor (TNF). On the contrary, treatment with Eubacterium rectum reduced TNF levels and lymphoma incidence rate in sensitized Eμ-Myc mice. In addition, lipopolysaccharide (LPS) in the resident microbiota of lymphoma patients and mice interacted with TNF signal, and enhanced the NF-kB pathway through MyD88-dependent TLR4 signal, which also enhanced the survival and proliferation of intestinal B cells. However, in the GI tract, a Eubacterium rectale-deficit gut microbiota stimulated B cells through enhanced TNF release and further sensitized B cells to LPS; these extracellular substances could bind to membrane TNFR1 and TLR4 and activated NF-kB signaling in a MyD88-dependent manner, as an extrinsic mechanism. These findings reveal the mechanism of inflammation-related lymphoma and the potential clinical principle of intestinal microbiota therapy targeting; In multiple myeloma, Xingxing Jian et al. found that nitrogen-recycling bacteria were significantly enriched in MM patients, such as Klebsiella pneumoniae. In addition, further single bacterial transplantation (wild-type Klebsiella pneumoniae and glnA-mutation Klebsiella pneumoniae) showed that the tumor burden of 5TGM1 mice transplanted with wild-type Klebsiella pneumoniae was higher. This study revealed a new mechanism of intestinal flora alteration in accelerating MM progression, suggesting that the intervention of intestinal flora in MM patients can be used as a new target for MM treatment. These studies showed that gut microbes are also closely related to the initiation and progression of hematological malignancies, but the relationship between the two needs more evidence to prove.

Point 2: 

The study found that intestinal flora can be used as a predictor of common adverse reactions associated with chemotherapy: it is not a new report. Also that the decrease of microbial during chemotherapy is related to the efficacy of chemotherapy.

Response 2: 

Thank you for this comment. Intestinal microbial abundance may be affected by chemotherapy (reference 9, 13, 17, 30, 37, and 46) and immunotherapy (reference 32). In contrast, intestinal microbial may affect chemotherapy and immunotherapy, TaChung Yu et al. reported that gut microorganisms and Fusobacterium nucleatum may alter the response of cancer cells to chemotherapeutics by mediating nucleotide synthesis and activating the autophagy pathway, respectively. Sophie Viaud et al. reported that Lactobacillus johnsonii and Enterococcus hirae translocate to the spleen and activate Th1 and Th17 immune responses, aiding cyclophosphamide treatment response. Evidence supports the role of the gut microbiome not only in cancer development and progression, but also in defining chemotherapeutic (5-fluorouracil, cyclophosphamide, irinotecan, oxaliplatin) and immunotherapeutic (anti-programmed death-ligand 1/anti-programmed cell death protein 1) effectiveness and toxicity. This evidence is supported by numerous in vitro, animal, and clinical studies that highlight the importance of microbial mechanisms in defining therapeutic responses. Therefore, the multiple effects of gut microbes on tumor characteristics provide a solid theoretical foundation for the development of personalized cancer treatment. However, if the microbiome is to be successfully translated into next-generation oncologic treatments, a new multimodal model of the oncomicrobiome must be conceptualized that incorporates gut microbial cometabolism of pharmacologic agents into cancer care.

Point 3:

The Authors mention (pag.5 and 7) "previous studies" and "relevant studies" without a correlation in the References: too vague.

Response 3:

Thank you for this comment. Correcting such errors and adding references have been done, and having accurate references improves the accuracy of our review. References were also presented as follows.

‘Previous studies’ is cited in references 19, 20 and 21; 'Relevant studies' is cited in reference 30.

Point 4:

The message reported in the paper is quite dangerous for a clinician because the study is interesting but the Conclusions not easy to be accepted.

Response 4:

Thank you for this comment. In the conclusion section, we have made some adjustments.  A detailed definition was inserted in the line 596-645 marked in red, which was also presented as follows.

Intestinal microbe as the second genome of human body, plays an important role in maintaining normal physiological functions in the human body, and hematological tumor progression and initiation are influenced by the balance of intestinal microbes. Therefore, based on published research, we may decrease the initiation of hematological malignancies and relieve the progression of hematological malignancies by regulating intestinal flora. A significant difference in intestinal microbial diversity exists between patients with hematological malignancies and healthy donors, additionally, in most patients of hematological malignancies, the microbes producing SCFAs are reduced or absent. The absence of probiotics and prebiotics may cause the excessive proliferation of pathogenic bacteria and intestinal microbial disorders, inflammation, even hematological tumor development. Significant epidemiological data have established that subjects with decreased SCFA levels have an increased risk of inflammatory disorders and cancer. It has been shown that SCFAs in the host gut and other parts can inhibit the growth and migration of cells, suppress histone deacetylases (HDAC), and induce apoptosis, thus reducing carcinogenesis extensively and providing future assistive strategies for the prevention and treatment of tumors. In leukemia, several studies reported that prebiotics or probiotics can restore intestinal barrier function, enhance the expression of intestinal antimicrobial proteins, promote immune response, increased insulin levels, and decreased plasma glucose concentrations, and further inhibit proliferation of leukemia cells and prolong survival. However, in certain situations, such as when the TET2 gene is deleted, translocation of probiotic Lactobacillus reuteri may lead to PMP, which can result in leukemia. In lymphoma, deficiency of probiotics Eubacterium recale may increase lymphoma incidence rates, while supplementation with it may reduce lymphomagenesis. And the supplementation of probiotic Lactobacillus johnsonii may reduce the incidence of lymphoma in cases of A-T. In multiple myeloma, sustained minimal residual disease negativity in multiple myeloma is associated with stool butyrate, and there is evidence that butyrate can induce apoptosis in MM cells. Additionally, probiotic Clostridium butyrate can inhibit the MM progression when supplemented. Therefore, functional foods like prebiotics (e. g. SCFAs) and probiotics (e. g. Clostridium butyrate) have beneficial influences in controlling and preventing cancer in preclinical and animal models.

Probiotics, such as Lactobacillus, may be effective in preventing infections in patients with suppressed immune systems. A phase I trial is studying the side effects and how well giving enteral nutrition, including Lactobacillus, works in preventing infections in patients undergoing donor stem cell transplant for hematologic cancer or myelodysplastic syndrome (NCT00946283). Another one investigates the effect of fermented milk supplementation on symptoms of disease and treatment in patients with multiple myeloma. Patients with multiple myeloma may experience symptoms related to the disease and/or treatment that affect the quality of life. Supplementing usual diet with a probiotic fermented milk product called kefir may contribute to reducing disease and treatment-related side effects through changing the intestinal bacteria community structure and related metabolism (NCT04530812). The clinical trial about butyrate which belongs to prebiotics, investigates whether the tolerance has been built for up to three 21-day cycles with Arginine Butyrate and ganciclovir/valganciclovir, or results in responses in patients with EBV (+) lymphoid malignancies (NCT00917826). This is an interesting approach since, in general, cancer survivors experience more rapid declines in health-related quality of life, including physical and psychological comorbidities. Therefore, prebiotics and probiotics are known to impact the composition and functional capacity of the microbiota with potential health benefits, which might be an adjuvant tool to rebalance the microbiota composition and improve inflammation and immune function, and cancer therapy will go beyond raising hope in the near future with the better understanding of cancer initiation and development.

Point 5:

Functional foods like prebiotics and probiotics have beneficial influences in controlling and preventing cancer in preclinical and animal models:in which way? What practical suggestion not too generic should be addressed to clinicians?

Response 5:

Thank you for pointing it out. As the second genome of the human body, the intestinal microbe plays an important role in maintaining normal physiological functions in the human body, and hematological tumor progression and initiation are influenced by the balance of intestinal microbes. Therefore, we may reduce the initiation of hematological malignancies and relieve the progression of hematological malignancies by regulating intestinal flora. A significant difference in intestinal microbial diversity exists between patients with hematological malignancies and healthy donors, additionally, in most patients with hematological malignancies, the microbes producing SCFAs are reduced or absent. The absence of probiotics and prebiotics may cause intestinal microbial disorders, excessive proliferation of pathogenic bacteria, inflammation, and hematological tumor development. Significant epidemiological data have established that subjects with decreased SCFA levels have an increased risk of inflammatory disorders and cancer. It has been shown that SCFAs in the host gut and other parts can inhibit the growth and migration of cells, suppress histone deacetylases (HDAC), and induce apoptosis, thus reducing carcinogenesis extensively, preventing and treating cancer. In leukemia, several studies reported that prebiotics or probiotics can restore intestinal barrier function, enhance the expression of intestinal antimicrobial proteins, promote immune response, increased insulin levels, and decreased plasma glucose concentrations, and further inhibit proliferation of leukemia cells and prolong survival. However, in certain situations, such as when the TET2 gene is deleted, translocation of probiotic Lactobacillus reuteri may lead to PMP, which can result in leukemia. In lymphoma, deficiency of probiotics Eubacterium recale may increase lymphoma incidence rates, while supplementation with it may reduce lymphomagenesis. And the supplementation of probiotic Lactobacillus johnsonii may reduce the incidence of lymphoma in cases of ataxia-telangiectasia (A-T). In multiple myeloma, sustained minimal residual disease negativity in multiple myeloma is associated with stool butyrate, and there is evidence that butyrate can induce apoptosis in MM cells. Additionally, probiotic Clostridium butyrate can inhibit the MM progression when supplemented. Therefore, functional foods like prebiotics (e. g. SCFAs) and probiotics (e. g. Clostridium butyrate) have beneficial influences in controlling and preventing cancer in animal and preclinical models.

Point 6:

Tables and figure are not easy for reading.

Response 6:

Thank you for this comment. We have already divided the image into three parts by adjusting the cite position , which indeed made our review more clear. A detailed definition was inserted in the line 255-258, line 436-439, and line 569-572 marked in red. We have corrected table 1-3, and a detailed definition was inserted in the line 179-180, line 390-391, and line 517-518 marked in red.

Reviewer 3 Report

The manuscript of Yinghong Zhu et al. is dedicated to discussing the possible role of gut microbiota in hematological cancer development, effectiveness of anticancer therapy, and the possibilities in the tumor course prediction. The focus of this literature review is on analysis of the relationships of populations of microbial cells in the intestinal tract with course of leukemia, lymphoma, and multiple myeloma. These aspects of the discussion emphasize the novelty and originality of the submitted manuscript. Indeed, “there are only a few publications summarizing the relationship between intestinal microbes and the initiation and progression of hematological malignancies”. The authors of this review consider in detail and summarize the published data on the topic of the article. Based on the performed analysis, the authors described the material in relevant paragraphs “Gut microbes …: clinical relevance”, “Intestinal microbial imbalance and the initiation and progression of …”, “The treatment prospect…”. The final part of the article “Conclusions” summarizes the reasonable arguments for the summary “…we can prevent the initiation of hematological malignancies and relieve the progression of hematological malignancies by regulating intestinal flora”.

The significance of this review is beyond doubt, since the presented summary of literature data and their analysis give an objective integral picture of existing approaches and ideas on the indicated topic and highlight possible ways of practical use of the accumulated knowledge.

In general the review is well presented; the data are of considerable novelty and interest.

I have some minor comments:

1. Tables 1-3 should be corrected for readability.

2. (line 201) When describing results obtained in mouse models, it is incorrect to use the term “patient”.

3. (line 307, etc.) It would be desirable to give a brief explanation regarding the “alpha diversity” and “beta diversity”.

4. The legend for Figure 1 should be directly below the figure.

5. The manuscript contains a number of inaccuracies and misprints in the text. Authors should find and correct the mistakes.

Author Response

Response to Reviewer 3 Comments

Point 1:

Tables 1-3 should be corrected for readability.

Response 1:

We greatly appreciate your positive comments. Specifically, we have corrected the Tables 1-3, and a detailed definition was inserted in the line 179-180, line 390-391, and line 517-518 marked in red.

Point 2: 

(line 201) When describing results obtained in mouse models, it is incorrect to use the term ‘patient’.

Response 2: 

Thank you for pointing it out, we have corrected the error.

Point 3:

(line 307, etc.) It would be desirable to give a brief explanation regarding the ‘alpha diversity’ and ‘beta diversity’.

Response 3:

Thank you for this comment, we have made some adjustments. A detailed definition was inserted in the line 129 and 145 marked in red, which was also presented as follows.

β-diversity: analyze the composition of microbial communities in different samples/groups; α-diversity: describe the abundance and diversity of microbial communities in a single sample.

Point 4:

The legend for Figure 1 should be directly below the figure.

Response 4:

Thank you for this comment, we have corrected the mistakes for Figure 1, and have already divided the image into three parts by adjusting the cite position , which indeed made our review more clear. A detailed definition was inserted in the line 255-258, line 436-439, and line 569-572 marked in red.

Point 5:

The manuscript contains a number of inaccuracies and misprints in the text. Authors should find and correct the mistakes.

Response 5:

Thank you for this comment, we have corrected the mistakes in the line 21-24, line 40-41, line 69-73, line 75, line 84-92, line 103-105, line 117, line 120, line 124-128, line 133-135, line 151-153, line 155-160, line 167-168, line 170-173, line 182, line 186-192, line 208, line 211-218, line 223-224, line 226-230, line 232-237, line 239, line 248, line 261, line 265-266, line 271, line 275-282, line 284-285, line 288, line 294-296, line 299-301, line 339-341, line 345, line 347-348, line 357-359, line 361-373, line 376-378, line 397-401, line 403-404, line 408-411, line 433-435, line 443-449, line 455-459, line 466, line 468-473, line 477, line 480-487, line 491-496, line 499-501, line 504-508, line 509-512, line 526-527, line 531-532, line 535-539, line 548-549, line 560-563, line 568-569, line 575, line 579-588, line 592, line 596, line 599-600, and line 670-671 marked in red.

Round 2

Reviewer 1 Report

The Authors thoroughly addressed the Reviewers' comments and made specific corrections that have significantly improved the revised manuscript. The clarifications they provided have enhanced the coherence and quality of the study. The manuscript’s clear and concise writing style contributes to the existing knowledge in its field. Therefore, the Reviewer suggests that it is suitable for publication.

Reviewer 2 Report

The Authors reviewed the paper in an excellent way answering deeply and correctly to the suggestions and criticisms of the reviewers.